# Construction of Polypyrrole-Coated CoSe_2_ Composite Material for Lithium-Sulfur Battery

**DOI:** 10.3390/nano13050865

**Published:** 2023-02-25

**Authors:** Yinbo Wu, Yaowei Feng, Xiulian Qiu, Fengming Ren, Jian Cen, Qingdian Chong, Ye Tian, Wei Yang

**Affiliations:** 1School of Automation, Guangdong Polytechnic Normal University, Guangzhou 510665, China; 2School of Chemistry and Chemical Engineering, Guangzhou University, Guangzhou 510006, China

**Keywords:** lithium-sulfur battery, transition metal selenides, metal-organic framework, polypyrrole, sulfur host

## Abstract

Lithium-sulfur batteries with high theoretical energy density and cheap cost can meet people’s need for efficient energy storage, and have become a focus of the research on lithium-ion batteries. However, owing to their poor conductivity and “shuttle effect”, lithium-sulfur batteries are difficult to commercialize. In order to solve this problem, herein a polyhedral hollow structure of cobalt selenide (CoSe_2_) was synthesized by a simple one-step carbonization and selenization method using metal-organic bone MOFs (ZIF-67) as template and precursor. CoSe_2_ is coated with conductive polymer polypyrrole (PPy) to settle the matter of poor electroconductibility of the composite and limit the outflow of polysulfide compounds. The prepared CoSe_2_@PPy-S composite cathode shows reversible capacities of 341 mAh g^−1^ at 3 C, and good cycle stability with a small capacity attenuation rate of 0.072% per cycle. The structure of CoSe_2_ can have certain adsorption and conversion effects on polysulfide compounds, increase the conductivity after coating PPy, and further enhance the electrochemical property of lithium-sulfur cathode material.

## 1. Introduction

At present, there is an urgent pursuit of energy storage equipment with high energy density and environmental friendliness, but the cathode material of lithium-ion batteries (LIBs) has nearly reached the limit of theoretical capacity in addition to having a high price [1,2,3,4]. The lithium-sulfur battery with high theoretical energy density (2600 Wh kg^−1^) and specific capacity (1650 mAh g^−1^) has become one of the promising secondary batteries for energy storage [5,6]. Nevertheless, the service life of the battery is bad and difficult to apply commercially because of the poor electroconductibility of sulfur and Li_2_S_2_, the “shuttle effect” of polysulfides, volume expansion, and other problems [7,8,9,10]. The structural design of carbon-based materials and the functional modification of electrode materials are used to improve electrochemical performance. The most effective design uses materials with high conductivity and strong chemical adsorption that are constructed into porous and loose structures, which are used as carrier compounds with the active substance S to prepare the positive electrode.

Researchers have conducted a lot of research on lithium-sulfur battery cathode materials. Non-polar porous carbon materials or polar porous carbon doped with B, N, O, P, S and other elements were used to load sulfur to improve the electrical conductivity of cathode materials [11,12,13,14,15]. For example, carbon nanotubes (CNTs) or carbon nanofibers (CNFs) [16], graphene carbon nanosheets [17], and porous carbons and their complexes [18] used to prepare sulfur composite cathode materials can significantly improve the cycle stability and rate performance of lithium-sulfur batteries. This improvement can be attributed to their high specific surface area, adjustable porous structure, and remarkable chemical/electrochemical stability. Cui et al. [19] found that metal sulfides alleviated the shuttle phenomenon by forming strong bonds with polysulfides and played a critical part in activating the electrochemical catalyst during charging, promoting the oxidation of Li_2_S back to S. Thereafter, various metal sulfides were used to capture polysulfides and promote the oxidation of Li2S into S in lithium batteries.

Similar to metal sulfide, selenides also have excellent catalytic activity and electrical conductivity as semimetals, indicating that metal selenides can relieve the “shuttle effect” of polysulfide by chemisorption and catalysis, thus improving the performance of S cathode [20]. Although the transition metal selenides exhibit polar interactions with lithium polysulfide or have effective catalytic effects on sulfur conversion, the extended cycle period is limited by serious volume expansion and poor conductivity in lithium-sulfur batteries. This hinders the process of large-scale application of the lithium-sulfur battery. To address these problems, the most effective strategies are combined with conductive carbon materials or conductive polymer coating designs. In contrast to ordinary carbon materials, conductive polymers do not need to be treated at high temperatures to be carbonized. Among many conductive polymers, polypyrrole (PPy) is the most widely used, showing excellent conductivity and a wide potential window. The PPy coating can enhance the electrical conductivity and lithium-ion diffusion of the cathode material while maintaining structural integrity. Jiang et al. [21] combined the characteristics of PPy with MOFs to increase the conductivity of the prepared material by five to seven orders of magnitude. Moreover, the PPy-MOF composite material constructed has appropriate ion channels to promote ion diffusion and transmission, thus achieving high rate performance.

In this article, the experimental preparation process is shown in Figure 1. We briefly discuss polyhedral hollow structure of selenide cobalt (CoSe_2_), which was synthesized by using metal-organic frameworks (MOFs) ZIF-67 as a template and precursor through high temperature selenization. The conductivity of Se is 1 × 10^−5^ S cm^−1^, which is 25 times higher than that of sulfur (5.0 × 10^−30^ S cm^−1^) [22]. Therefore, charge transfer kinetics and overall electrochemical performance can be improved. ZIF-67 was chemically modified to modify the active site for physical adsorption and impregnation of sulfur and polysulfide. The Lewis CoSe_2_ synthesized has an appropriated electron structure and catalytic activity, and reduces the concentration of polysulfide in electrolyte through chemical capture and concentration of sulfide, to facilitate the dynamics of the oxidation reduction of sulfur transformation [23,24]. CoSe_2_ coated with PPy increases the conductivity of the cathode materials, obtain a considerable specific capacity and a long cycle life at 3 C, and provides an experimental basis for solving the sulfur cathode problem of a lithium-sulfur battery.

## 2. Experimental Section

### 2.1. Preparation of ZIF-67 and Hollow CoSe_2_

The CoSe_2_ samples were prepared by the combination of co-precipitation and high- temperature carbonization. First of all, according to the stoichiometric ratio, 8 mmoL cobalt nitrate (Co(NO_3_)_2_•6H_2_O, AR, Shanghai Maclin Biochemical Technology Co., Ltd., Shanghai, China) and 32 mmoL 2-methylimidazole (AR, Shanghai Maclin Biochemical Technology Co., Ltd., Shanghai, China) were each added to a beaker containing 100 mL methanol solution (99.9%, AR, Aladdin Chemistry Co., Ltd. Shanghai, China), stirring until dissolved. The two solutions were mixed and stirred for 30 min, and then left to rest at room temperature for 24 h. The purple precipitation was washed by methanol, centrifuged, and dried at 80 °C for 12 h to get purple powder ZIF-67. ZIF-67 and selenium powder (AR, Aladdin Chemistry Co., Ltd., Shanghai, China) were placed in a crucible with a certain mass ratio (1:2) and calcined at 600 °C for 3 h in a tubular furnace in nitrogen atmosphere to get CoSe_2_ samples.

### 2.2. Preparation of Hollow CoSe_2_@PPy Dodecahedrons

First, 1.5 mmol sodium p-toluene sulfonate (p-TSS, AR, Shanghai Maclin Biochemical Technology Co., Ltd. Shanghai, China) was added to 30 mL of mixed solution containing pure water/ethanol (*v*/*v*, 1:1), followed by 0.116 g of pyrrole monomer (99.9%, AR, Aladdin Chemistry Co., Ltd. Shanghai, China), and stirred until a uniform mixed solution A. At the same time, 3.75 mmol ammonium persulfate (APS, AR, Shanghai Maclin Biochemical Technology Co., Ltd. Shanghai, China) as an oxidant was added to 30 mL aqueous solution after intense agitation to gain solution B. Then 0.1 g CoSe_2_ powder was placed into a clean beaker with 40 mL DI and ultrasonic dispersed for 30 min, and then solution A was slowly dropped into the beaker. Solution B was also dropped 30 min later and polymerized under ice bath conditions. The sample was then placed in darkness for 24 h, and the residue was successively removed with pure water and methanol. At last, the black sample was dried overnight at 60 °C to gain CoSe_2_ coated with in situ functional layers modified by PPy.

### 2.3. Preparation of Hollow CoSe_2_@PPy-S

The CoSe_2_@PPy nanocomposites were mixed with S at 3:7 mass ratios and sealed in glass tubes. After being heated at 155 °C for 12 h, the products were collected for characterization after cooling. CoSe_2_-S material was prepared under the same conditions for comparison.

### 2.4. Measurement of Material Characteristics

The crystal phase of the sample was analyzed by XRD (PW3040/60, Eindhoven, The Netherlands). The morphological and structural characteristics of the prepared materials were investigated by SEM (JSM-7001F, Tokyo, Japan), and the lattice fringes of materials were tested by HRTEM (JEM-2100F, Tokyo, Japan). The PPy was further detected by FTIR (Spectrum100, Waltham, MA, USA). TheA specific surface area and pore size of the cathode matrix materials were measured and analyzed by Means of ASAP 2460(Norcross, GA, USA) BET specific surface analyzer. Sulfur mass was calculated by TGA (NETZSCH STA 449 F3/F5, Germany) in nitrogen at 10 °C min^−1^ from 30 to 600 °C. The chemical composition and valence of the composites were determined by XPS (Thermo Scientific K-Alpha, Waltham, MA, USA). The adsorption performance of the composite material to lithium polysulfide was tested by UV-vis spectrometer (UV-3600, Kyoto, Japan).

### 2.5. Electrochemical Characterization of Materials

The positive electrode cut piece was made into a circular piece with a diameter of 12 mm for use. Lithium plates (thickness of 2 mm) were used for the negative electrode, Celgard2400 for the diaphragm, and 1.0 mol/L^−1^ LiTFSI-DOL/DME (*v*/*v*, 1:1) −0.1 M LiNO_3_ for the electrolyte (40 L/per cell). The test button battery (CR2032) was assembled in a glove box filled with argon. The resulting battery tests were performed on a charge-discharge meter at room temperature, alternating current impedance test and cyclic voltammetry (CV) tests were experimented at the CHI600 electrochemical workstation.

### 2.6. Visualization Experiment Test

Firstly, to form a concentration of 0.05 M Li_2_S_6_ solution, the sulfur powder and lithium disulfide (molar ratio 5:1) were added to DOL/DME(*v*/*v*,1:1) solution and whisked together at 60 °C for 48 h until thoroughly combined to gain Li_2_S_6_ solution for reserve. Then 20 mg of the active substance was added to 5 mL of diluted (0.005 M) Li_2_S_6_ solution and let to stand for 6 h to observe the change in color of the solution. Then the supernatant fluid was taken for the UV-vis absorption spectrum test.

## 3. Results and Discussion

The characteristic peaks of the prepared ZIF-67 (Appendix A) are sharp, indicating that it has good crystallizability. The elemental sulfur is a rhombic crystal system bonded by a covalent sulfur constituent structure, corresponding to the standard card of S_8_ (PDF#89-2600) in Appendix A. At present, CoSe_2_ with a polyhedral porous structure has been successfully synthesized in the experiment, and its XRD is shown in Figure 2a. CoSe_2_ corresponds to the standard cards of CoSe_2_ (PDF#65-3327, cubic structure; PDF#53-0449, rhombic structure) and the Pa-3 space group. The cubic phase selenides have better catalytic capacity than the orthogonal phase. However, the periodic arrangement of atoms at the intersecting interfaces of different crystal types will change dramatically, and the atoms have higher degrees of freedom, resulting in defects in the crystal plane that can act as catalytic sites. CoSe_2_ with higher catalytic activity can be obtained by the simultaneous growth of the two kinds of crystal, which can produce more mixed grain boundary planes. The XRD patterns of CoSe_2_, S, PPy, CoSe_2_@PPy, CoSe_2_@PPy-S, and CoSe_2_-S composites are shown in Figure 2b. PPy shows a very wide peak at 20–30°, corresponding to the impalpable structure of PPy [25]. When the surface of CoSe_2_ material is coated with PPy, CoSe_2_@PPy composite material shows the characteristic peak of CoSe_2_ without the miscellaneous peak. The results showed that the CoSe_2_ crystals in CoSe_2_@PPy composites did not change after pyrrole polymerization. Because of the amorphous structure and high dispersion of PPy, it has no obvious performance in composites. When loaded with sulfur, the intensity of the diffraction peak of CoSe_2_@PPy-S becomes relatively weak, and a sharp sulfur diffraction peak appears at the same time, showing that sulfur exists together well with CoSe_2_@PPy composite in the crystal structure. 

Figure 2c shows the N_2_ adsorption/desorption curve. The surface area of CoSe_2_ and CoSe_2_@PPy materials is 207.55 and 151.78 m^2^ g^−1^, respectively, by BET calculation, and the pore diameter distribution ranges from 2 to 50 nm, among which the main pore size distribution is 3.9 and 4.9 nm, respectively. The pore volume of CoSe_2_ and CoSe_2_@PPy materials is 0.296 and 0.218 cm^3^ g^−1^, respectively, by BJH method. The surface pore structure of CoSe_2_ and CoSe_2_@PPy materials provides ample space for sulfur storage. At the same time, it makes it easier for electrolyte seepage material to have better contact with active substances, to speed up the charge and discharge ion transport process, making the active substances in the cathode material reacted more sufficient, improving the utilization rate of sulfur. Figure 2d shows the thermogravimetric curve of CoSe_2_@PPy-S and CoSe_2_-S under nitrogen protection at 30–600 °C. Significant mass loss occurs at 150–300 °C. According to the thermogravimetric curve of the material, the loss within this temperature range is mainly due to the evaporation of sulfur, which accounts for about 70% [26]. It is consistent with the mass proportion of sulfur in the substrate. Weight loss temperature of CoSe_2_-S is significantly lower than that of CoSe_2_@PPy-S.

Figure 3 displays the infrared spectrum analysis. The characteristic peaks at 1305 and 1046 cm^−1^ are due to the presence of vibration peaks for the =C–H plane of the PPy long chain, and the absorption peaks at 1457 and 1559 cm^−1^ are derived from the basic vibration peaks of the pyrrole ring [26,27,28,29]. In addition, the stretching vibration peak of C–N and the out-of-plane vibration peak of =C–H is situated at 1119 cm^−1^ and 914 cm^−1^, respectively. From the infrared spectra of PPy and CoSe_2_@PPy-S composites, it is obvious that when elemental sulfur is heated and melted into CoSe_2_@PPy materials, CoSe_2_@PPy-S composites still show the typical characteristic peak of PPy. The results show that the structure of PPy in CoSe_2_@PPy-S composites does not change after the sulfur melting reaction, and PPy still exists in CoSe_2_@PPy-S composites. However, the content of PPy in CoSe_2_@PPy-S composite is small and relatively dispersed, so the peak value of the composite in the infrared spectrum is much lower than that of pure PPy.

SEM results in Figure 4a show the ZIF-67 material has a typical rhomboid dodecahedron structure, and the synthesized product has a smooth surface, uniform particle size, and good dispersion. Figure 4b is the scanning electron microscope image of CoSe_2_. It can be seen that the polyhedron structure of the ZIF-67 precursor can be maintained after further reflow calcination, but the surface of the polyhedron becomes rough and consists of many fine particles. The CoSe_2_@PPy material coated with conductive polymer PPy is shown in Figure 4c. The surface is coated with many irregulars and heterogeneous PPy particles. From the broken CoSe_2_@PPy particle on the right side, it can be seen that the structure is still hollow, and the existence of the inner hollow part provides enough storage space for the sulfur elements. Figure 4f,g corresponds to the HR-TEM image of CoSe_2_ composite material. In the figure, the crystal plane spacing of 0.332 nm, 0.269 nm, and 0.234 nm correspond to the (111), (210), and (211) crystal planes of CoSe_2_ crystal, respectively. Therefore, it further indicates that CoSe_2_ crystal exists in the composite, and this result corresponds well to the XRD pattern of the material. Figure 4h,i shows TEM images of PPy coated on CoSe_2_ surface. The polymerization of pyrrole is not uniform along the surface of the matrix, and the coating thickness is between 30–50 nm. These PPy particles with inconsistent particle size can well cover the matrix material, while the interior of the material is still hollow structure.

Figure 4e is the SEM picture of CoSe_2_@PPy-S composite. The basic morphology of CoSe_2_@PPy was not changed after the addition of sulfur element, and no large sulfur element was found on the surface of CoSe_2_@PPy-S composite. It indicates that sulfur element is well distributed in the internal part of CoSe_2_@PPy material, so the active substance can be well fixed in the material, decreasing sulfur loss during reaction and improving the electrochemical stability of the battery. The existence and specific distribution of each element in the CoSe_2_@PPy-S composite is further determined by elemental energy spectrum analysis, as shown in Figure 4j. Among them, Co, Se and N elements come from CoSe_2_@PPy materials. The symmetrical distribution of sulfur element in the composite shows that sulfur is well combined with CoSe_2_@PPy material and is also conducive to a more adequate reaction.

Figure 5a shows the full spectrum of XPS of the CoSe_2_@PPy-S composite, from which the existence of Co, Se, N, C, and S elements can be seen. In Se 3d spectrograms (Figure 5c), the fitting peaks at 54.9 and 55.7 eV are ascribed to the presence of Se 2d3/2 and Se 3d5/2, respectively. In addition, two peaks appear at 58.2 eV and 59.5 eV, which are consistent with the Se-O bond [30]. In the spectrum of Co 2p (Figure 5b), at 781.2 and 797.1 eV, there are two wide peaks related to Co 2p3/2 and Co 2p1/2, respectively [31,32,33]. The characteristic peaks of Co 2p and Se 3d were found in XPS spectra, which further confirmed the existence of CoSe_2_ in the composites, corresponding well with XRD spectra. Figure 5d is the XPS spectra of CoSe_2_@PPy-S composite S 2p, which mainly contains six fitting peaks, among which two fitting peaks located at 163.5 and 164.5 eV pertain to S 2p3/2 and S 2p1/2, respectively. The fitting peak of binding energy at 166.1 eV and 164.7eV are due to the presence of S–O [27]. As shown in Appendix A, the fitting peak at 162.4 eV is derived from sulfide, from which it could be inferred that sulfur could interact with CoSe_2_ [33]. In addition, a wide fitting peak at 168.8 eV indicates the existence of sulfate in the composite. In Figure 5e, C 1s of the composite material is divided into four main fitting peaks at 284.6, 285.8, 286.5, and 288.6 eV, belonging to C–C/C=C, C–N, C–O and C=O, respectively [26]. Figure 5f shows the XPS spectra of N 1s. N 1s is divided into three peaks corresponding to pyridine type nitrogen, pyrrole type nitrogen and graphite type nitrogen, and their binding energies are 398.5, 399.6 and 401.3 eV, respectively [26,34,35]. The N element in the material mainly comes from polypyrrole, so it can be concluded that the polypyrrole exists in the material. 

Figure 6a presents the CV curve of the CoSe_2_@PPy-S composite material. The two reduction peaks near 2.32 and 2.04 V correspond to the transformation of S_8_ into long-chain lithium polysulfides and then into short-chain Li_2_S_2_/Li_2_S. The oxidation peak at 2.43 V is the conversion of Li_2_S_2_/Li_2_S to lithium polysulfides and finally to S_8_ [36]. Compared with bare sulfur and CoSe_2_-S composites (revealing in Appendix A), CoSe_2_@PPy-S has a larger peak current and peak area in CV curves. The peak area does not change significantly after cycling, indicating that CoSe_2_@PPy-S composite cathode materials have good cycle stability. CoSe_2_ material prevents the disulfide from dissolving in the organic electrolyte and transferring back and forth migration between electrodes. The existence of PPy has greatly improved the conductivity of the cathode material.

Figure 6b reveals the charge–discharge curve of bare sulfur, CoSe_2_-S, and CoSe_2_@PPy-S composites at a 0.05 C rate. The composite electrode has two discharge platforms and one charging platform, corresponding well to the CV curves of the three materials. Compared with bare sulfur and CoSe_2_-S electrodes, the charge and discharge platforms of CoSe_2_@PPy-S composite are more stable, indicating that the capacity of sulfur is fully developed. In the meantime, the dropout voltage between the charge and discharge curves of the CoSe_2_@PPy-S composite electrode (ΔE = 227 mV) is significantly smaller than that of CoSe_2_-S (ΔE = 233 mV) and bare sulfur electrode (ΔE = 270 mV), indicating that the CoSe_2_@PPy-S composite electrode appears small polarization. 

Figure 6c shows the cycle performance curves of bare sulfur, CoSe_2_-S, and CoSe_2_@PPy-S at 0.2 C. Appendix A shows that CoSe_2_ and CoSe_2_@PPy materials provide little capacity. The initial discharge capacity of the CoSe_2_@PPy-S composite electrode is 690 mAh g^−1^, and its initial coulomb efficiency is 96.5% at 0.2 C, while the CoSe_2_-S and bare sulfur electrode is 659 and 433 mAh g^−1^, respectively. After 200 cycles, the discharge capacities of CoSe_2_-S and bare sulfur electrodes attenuate to 288 and 238 mAh g^−1^, respectively. The CoSe_2_@PPy-S composite electrode retains a specific discharge capacity of 376 mAh g^−1^ and a coulomb efficiency of 99.2%, due to the chemisorption of metal oxides to lithium polysulfide and good conductivity of PPy. Moreover, Figure 6f shows the electrochemical performance of CoSe_2_@PPy-S up to 200 cycles at 3 C. The discharge capacity of CoSe_2_@PPy-S composite cathode can reach 341 mAh g^−1^. After 200 cycles, the reversible capacity of 292 mAh g^−1^ can still be maintained, and the capacity decrease of each cycle is only 0.072%.

Figure 6d states the rate performance of bare sulfur, CoSe_2_-S, and CoSe_2_@PPy-S. CoSe_2_@PPy-S composite electrodes provide high discharge capacities of 1003, 756, 529, 445, 375, 307, and 128 mAh g^−1^ at 0.05, 0.1, 0.2, 0.5, 1, 2, and 5 C, respectively. The discharge capacity of CoSe_2_@PPy-S composite electrode can yield 531 mAh g^−1^ when returning to 0.1 C again. The charge and discharge capacities of CoSe_2_@PPy-S electrode at different rates are significantly higher than those of CoSe_2_-S and bare sulfur electrode, on account of the strong intermolecular interaction between CoSe_2_ and lithium polysulfide, and the CoSe_2_ structure can supply ample active sites to adsorb and catalyze lithium polysulfide. Moreover, conductive PPy improves the electronic conductivity of CoSe_2_@PPy-S composites and accelerates the electrochemical reaction kinetics.

Figure 6e shows the charge–discharge curves of CoSe_2_@PPy-S composite electrode at different rates. Similar to bare sulfur and CoSe_2_-S (Appendix A), the CoSe_2_@PPy-S composite electrode shows the highest initial capacity of 1003 mAh g^−1^, while CoSe_2_-S and bare sulfur electrodes are 955 and 608 mAh g^−1^, respectively. Meanwhile, the capacity retention of CoSe_2_@PPy-S composite decreases slowly. It indicates that the CoSe_2_@PPy-S composite has the best electrochemical performance. 

In order to further explore the electrochemical capacity performance of CoSe_2_@PPy-S, Figure 7a shows that the CV curves were tested in the range of 0.1~1.0 mV s^−1^ and maintained similar shapes at different scanning rates, indicating that the electrode has nice cyclic reversibility in the process of lithiation/delithiation. The capacitance contribution of the electrode material can be qualitatively analyzed using the following two equations [37].
(1)$$i=av^{b}$$
(2)log(i)=blog(v)+log(a) 

In the CV test, the corresponding peak current values (*i*, mA) were conducted at different voltage scanning rates (*v*, mV s^−1^). The b value was calculated according to the above formula to determine whether the electrode material belongs to the diffusion behavior or pseudocapacitance behavior during charge and discharge. The result of b value is 0.5, and the electrode material behaves as a battery type. If the b value is 1, it is considered to demonstrate capacitive charge storage behavior [33,37]. In Figure 7b, corresponding b value fitting results of the CoSe_2_@PPy-S electrode show that b values of peak 1 and peak 3 are 0.765 and 0.654, respectively, which are a mixture of capacitive type and diffusion type. Figure 7c shows the capacitance contribution ratio of the CoSe_2_@PPy-S electrode reaches 94.1% at 1 mV s^−1^, higher than 87.9% of CoSe_2_-S electrode (Appendix A), implying that the PPy layer coated on the material is beneficial to the transmission of Li^+^ and improves the discharge capacity of the composite [34,35].

The Randles–Sevcik Equation (3) was used to calculate the CV data of CoSe_2_@PPy-S composites at different scanning rates, and the lithium-ion diffusion coefficient was obtained [38]. The peak current (*I_p_*) varies linearly with the square of the scanning rate (*v*^1/2^):(3)$$I_{p}=2.69\times 10^{5}n^{3/2}AD_{{\mathrm{Li}}^{+}}^{1/2}C_{{\mathrm{Li}}^{+}}v^{1/2}\;$$
where *n* is the number of electrons in the reaction process (*n* = 2), *A* is the electrode area (*A* = 1.13 cm^2^), and *C*_Li_^+^ is the Li^+^ concentration in the electrode (*C*_Li_^+^ = 1.0 × 10^−3^ mol cm^−3^). The *D*_Li_^+^ values calculated from peak currents of peaks 1, 2, and 3 are 4.801 × 10^−12^, 0.559 × 10^−12^, and 0.414 × 10^−12^ cm^2^ s^−1^, respectively. CoSe_2_@PPy can accelerate the mobility of electrons, and ions in the sulfur electrode restrict the shuttle effect of lithium sulfide, optimizing the overall electrochemical performance of lithium-sulfur battery.

A symmetrical cell containing Li_2_S_6_ electrolyte was assembled, and the reaction kinetics were studied with CV. As shown in Figure 8a, CoSe_2_@PPy-L_2_S_6_ electrode has a higher redox current, which indicates that CoSe_2_@PPy is beneficial for accelerating the conversion reaction of lithium polysulfide. To further study the adsorption performance of various materials for lithium polysulfide, we conducted UV-vis absorption spectrum analysis on super P, PPy, CoSe_2,_ and CoSe_2_@PPy composite materials, and the test outcome is displayed in Figure 8b. The peak at 416 nm in the figure is related to S_6_^2-^, indicating the presence of Li_2_S_6_ in solution [31,33]. After four materials were added to the polysulfide solution and left standing for 24 h, the absorption peak intensity of CoSe_2_ and CoSe_2_@PPy materials weakened greatly at 416 nm, because CoSe_2_ and CoSe_2_@PPy had a strong interaction with Li_2_S_6_, so that they could better adsorb Li_2_S_6_. The illustrations in the digital image also confirm this. 

In Figure 9, the impedance spectra of the CoSe_2_@PPy-S, CoSe_2_-S, and bare sulfur electrode materials before circulation are all composed of semicircles in the high-frequency region and diagonal lines in the low-frequency region. The equivalent circuit diagram was constructed according to a Nyquist curve, and the resistance value was obtained by fitting the EIS data with Zview software. The semicircle part is connected with the charge transfer resistance (Rct), corresponding to the interfacial capacitance (CPE), while the slope line is the Warburg impedance (Zw), attributed to the diffusion ability of lithium-ion in the battery [34,35]. The Rct of the bare sulfur and CoSe_2_-S electrodes before cycling were 30 and 56 Ω, respectively. Compared with these two samples, the AC impedance diagrams of the composite electrode showed a smaller resistance of 29 Ω, indicating that the CoSe_2_@PPy-S composite has a lower charge transfer barrier and better conductivity of the battery.

## 4. Conclusion

A CoSe_2_@PPy-S composite with a special structure was prepared by a simple three-step reaction and has remarkable cycling stability and rate performance. The discharge capacity of CoSe_2_@PPy-S composite electrode is 341 mAh g^−1^ and low capacity decay rate of 0.072% after 200 cycles at 3 C. A rich active site exists in CoSe_2_ materials that can chemically anchor sulfide. The shell parts can form a physical barrier to further limit lithium polysulfide inside the cathode material, making it hard for polysulfide to dissolve so that diffusion occurs in the electrolyte. CoSe_2_@PPy has good electrical conductivity, promotes the electron migration in the reaction process, and speeds up the redox reaction. At the same time, its fluffy structure can also adsorb lithium polysulfide well and further improve the electrochemical performance of the electrode. The article provides a simple and new thought: to prepare high performance cathode materials for lithium-sulfur batteries.

## Figures and Tables

**Figure 1 nanomaterials-13-00865-f001:**
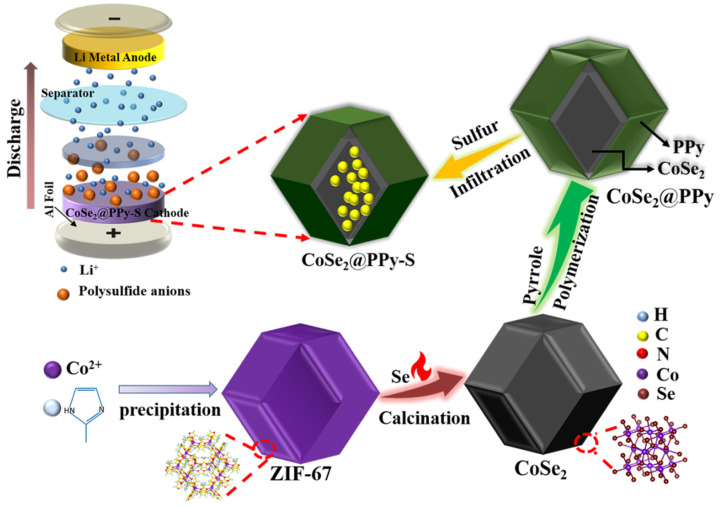
Flow chart of the preparation of CoSe_2_@PPy-S composites.

**Figure 2 nanomaterials-13-00865-f002:**
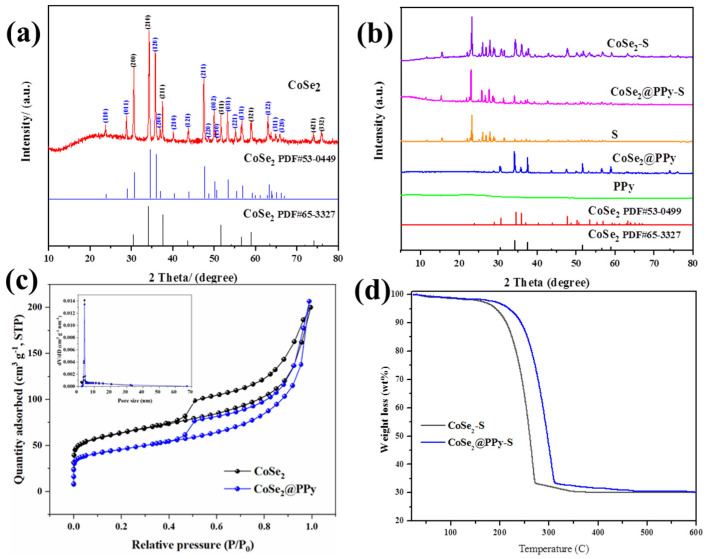
(**a**) XRD patterns of CoSe_2_ composites. (**b**) XRD patterns of PPy, CoSe_2_@PPy, S, CoSe_2_@PPy-S, and CoSe_2_-S composites. (**c**) The N_2_ adsorption/desorption curves of CoSe_2_ and CoSe_2_@PPy composites. The illustration is the aperture distribution of CoSe_2_ and CoSe_2_@PPy composites. (**d**) TGA curve of CoSe_2_@PPy-S composites.

**Figure 3 nanomaterials-13-00865-f003:**
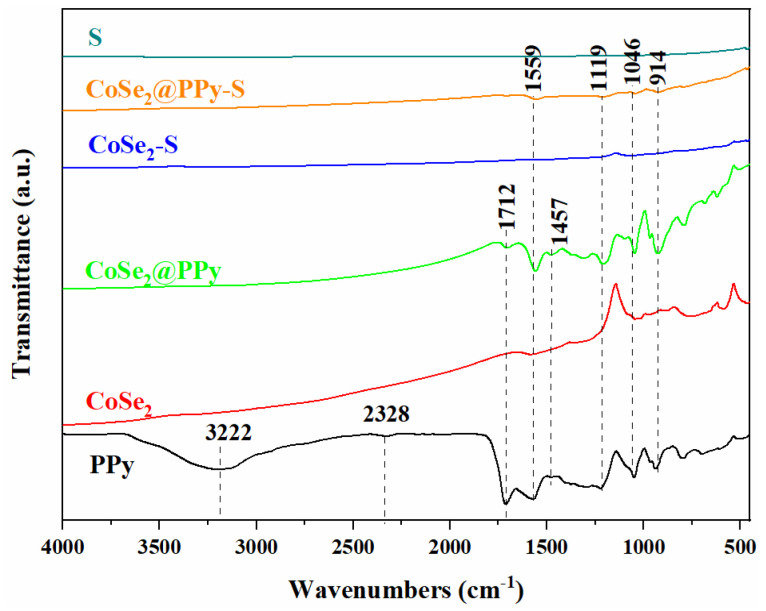
FTIR spectra of PPy, CoSe_2_, CoSe_2_@PPy, S, CoSe_2_@PPy-S, and CoSe_2_-S composites.

**Figure 4 nanomaterials-13-00865-f004:**
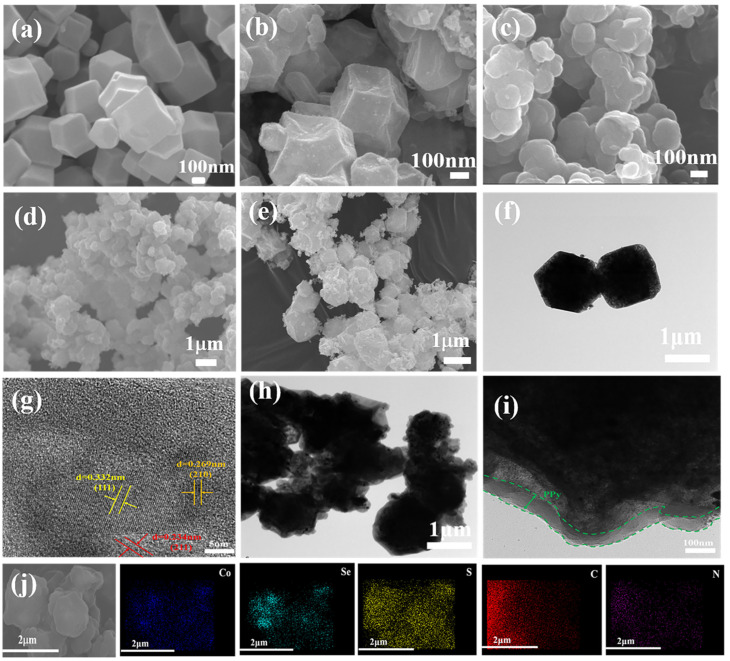
SEM images of (**a**) ZIF/67, (**b**) CoSe_2_, (**c**) CoSe_2_@PPy, (**d**) CoSe_2_-S, (**e**) CoSe_2_@PPy-S, HR-TEM images of (**f**,**g**) CoSe_2_, (**h**,**i**) CoSe_2_@PPy, and (**j**) elemental mapping results of Co, Se, C, N and S in CoSe_2_@PPy-S composites.

**Figure 5 nanomaterials-13-00865-f005:**
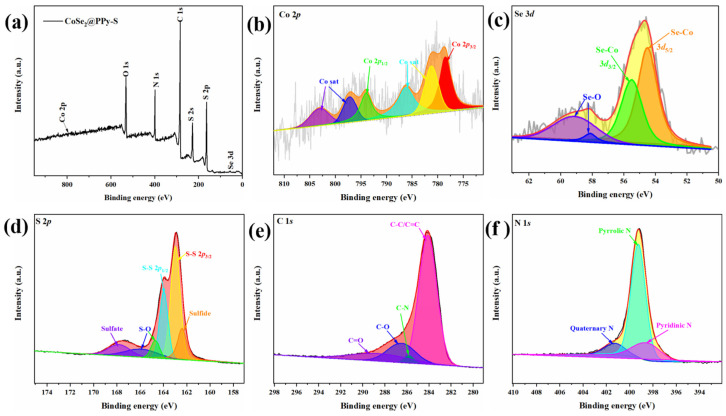
XPS spectra of CoSe_2_@PPy-S (**a**) survey spectrum, (**b**) Co 2p XPS spectrum, (**c**) Se 3d XPS spectrum, (**d**) S 2p XPS spectrum, (**e**) C 1s XPS spectrum, and (**f**) N 1s XPS spectrum.

**Figure 6 nanomaterials-13-00865-f006:**
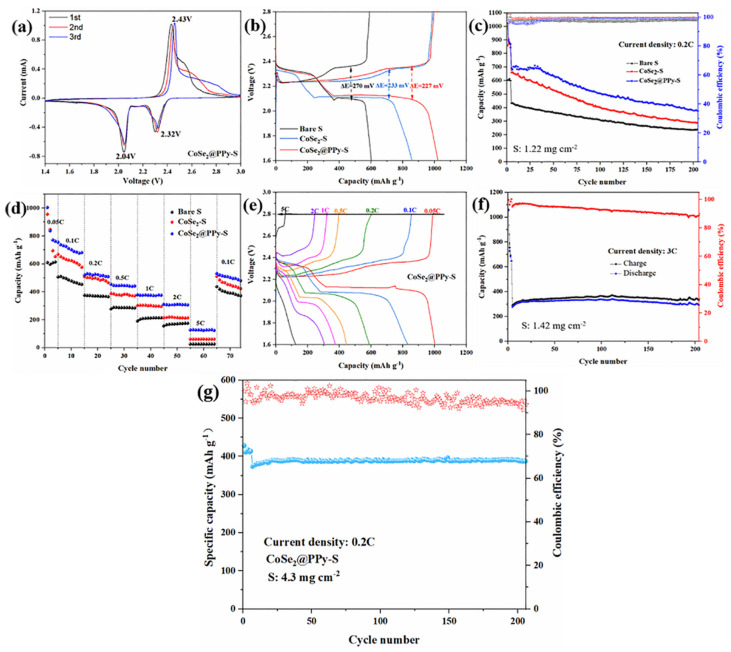
(**a**) CV curves of CoSe_2_@PPy-S composite at 0.1 mV s^−1^. (**b**) The comparison of discharge/charge curves of bare sulfur, CoSe_2_-S, and CoSe_2_@PPy-S cathodes. (**c**) Cyclic properties of bare sulfur, CoSe_2_-S, and CoSe_2_@PPy-S composites at 0.2 C after three cycles of activation at 0.05 C. (**d**) Rate capacity of bare sulfur, CoSe_2_-S, and CoSe_2_@PPy-S composite. (**e**) Charge and discharge curves of CoSe_2_@PPy-S electrode composites at different current densities. (**f**) Cycling performance of CoSe_2_@PPy-S at 3 C over 200 cycles. (**g**) Cycling performance of CoSe_2_@PPy-S at 0.2 C.

**Figure 7 nanomaterials-13-00865-f007:**
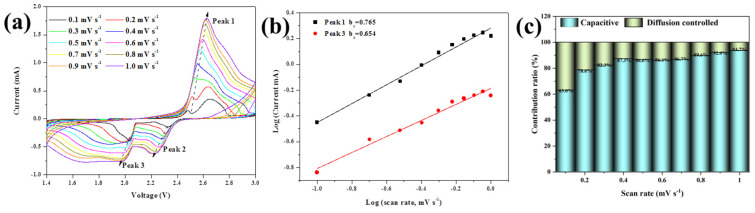
(**a**) CV curves of CoSe_2_@PPy-S composites at different scanning rates. (**b**) Profile of log(*i*) versus log(*v*) plots. (**c**) The charge contribution ratio of capacitance and diffusion at various scan rates.

**Figure 8 nanomaterials-13-00865-f008:**
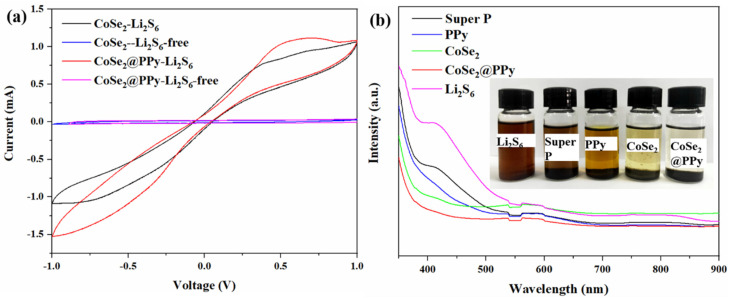
(**a**) CV curves of CoSe_2_ and CoSe_2_@PPy composites between −1.0 and 1.0 V at a potential sweep rate of 5 mV s^−1^. (**b**) UV–vis spectra and adsorption photographs of Li_2_S_6_ solution with blank, super P, PPy, CoSe_2_, and CoSe_2_@PPy composites.

**Figure 9 nanomaterials-13-00865-f009:**
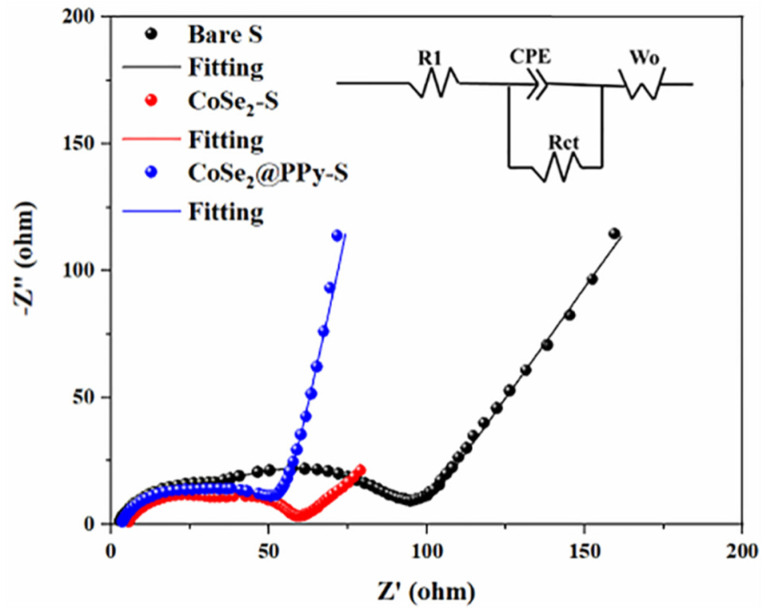
EIS spectra of bare sulfur, CoSe_2_-S, and CoSe_2_@PPy-S electrodes before cycling.

## Data Availability

The data presented in this study are available on request from the corresponding author.

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
