# Peer review of "Construction of Polypyrrole-Coated CoSe2 Composite Material for Lithium-Sulfur Battery"

_nanomaterials, 2023, doi:10.3390/nano13050865_

Round 1

Reviewer 1 Report

In their manuscript Dr Yinbo Wu et al. described the fabrication of a CoSe  structure embeded with PPyr film to support the lithiation and delithiation process. They characterized the material by several technics and studied the charge and discharge process. First of all I think that this study is quite interesting because of the use of hybrid material is a great advantage in many field and expecially in energy storage. However there are a lot of mistakes and errors and English should be corrected. Thus it is recommended that this manuscript be accepted for publication in Nanomaterials once the authors revise it by addressing the following major points:

- My major question concerns the XPS analysis of the CoSe2@PPy-S. The signals of Co and Se are very low while N or C signals are huge. In my opinion the insertion of S destroy all the previous CoSe2 structure. The author should comment or give us some explanation. The absence of CoSe questions the usefulness of the study. 

- For all electrochemical test, the authors should specify the composition of used solution.

- From XPS spectrum, authors stated that N comes from pyrrole. It should be possible however authors should try to quantify different ratios (i.e N/C, N/S or other) before statement.

- XPS of CoSe2 alone should be provided for comparison

- Some conclusion are very preliminar and not supported by experimental data. for example, fig 8.a did not show a real improvement with PPy. authors mentioned current density but it is current intensity.

- some references are needed in the introduction to support some statement.

Author Response

Dear reviewer:

Thank you for your decision and constructive comments on my manuscript. We have carefully considered the suggestions of the Reviewers. We have tried our best to improve and made some changes to the manuscript.

The yellow part has been revised according to your comments. Revision notes, point-to-point, are given as follows:

  1. Comment: My major question concerns the XPS analysis of the CoSe2@PPy-S. The signals of Co and Se are very low while N or C signals are huge. In my opinion the insertion of S destroy all the previous CoSe2 structure. The author should comment or give us some explanation. The absence of CoSe questions the usefulness of the study.

Response: Thank you very much for your careful review and constructive suggestions with regard to our manuscript. It might be a mass of sulfur load in the interface of cobalt selenide which leads the spectrum peaks of Se and Co are very weak. On the other hand, polypyrrole contains a lot of nitrogen, N element on the surface of CoSe2 increases after PPy coating. As a result the peak spectrum of N shows stronger.

  1. Comment: For all electrochemical test, the authors should specify the composition of used solution.

Response: In the previous submission, we put the experimental parts such as sample preparation and test conditions in the Supplementary material. Now we've put them in the manuscript and refined the details. The composition and concentration of electrolyte, the ratio of electrolyte-to-sulfur, and other electrochemical test parameters are listed in the revised manuscript. Thank you very much for your careful review.

  1. Comment: From XPS spectrum, authors stated that N comes from pyrrole. It should be possible however authors should try to quantify different ratios (i.e N/C, N/S or other) before statement.

Response: Thank you very much for your careful review and constructive suggestions with regard to our manuscript. According to the result of XPS testing, the atomic ratio of N/C is 18.33%, the atomic ratio of N/S is 45.04%.

  1. Comment: XPS of CoSe2 alone should be provided for comparison

Response: Thank you very much for your careful review and constructive suggestions with regard to our manuscript. As shown in Fig. S7, XPS of CoSe2 have been provided in the Supplementary material of the revised manuscript.

Fig. S7. XPS spectra of CoSe2 (a) Co 2p XPS spectrum, (b) Se 3d XPS spectrum, (c) survey spectrum

  1. Comment: Some conclusions are very preliminar and not supported by experimental data. for example, fig 8.a did not show a real improvement with PPy. Authors mentioned current density but it is current intensity.

Response: Thank you very much for your careful review and constructive suggestions with regard to our manuscript. As shown in Fig. 8a, CoSe2@PPy-L2S6 electrode has the highest redox current, which indicates that CoSe2@PPy is beneficial to accelerate the conversion reaction of lithium polysulfide. Since all of our electrodes are the same size, we describe as current density for current here. We have revised this description in the revised manuscript.

  1. Comment: some references are needed in the introduction to support some statement.

Response: Thank you very much for your careful review and constructive suggestions with regard to our manuscript. We have updated the references.

Reviewer 2 Report

Summary:

This Research Article nanomaterials-2226659 titled, “Construction of polypyrrole coated CoSe2 composite material for lithium-sulfur battery,” reports the synthesis of a hollow structure of CoSe2 with a surface coating of conductive PPy. The resulting CoSe2@PPy is then used as a cathode host to prepare a CoSe2@PPy-S composite cathode, with CoSe2 to keep the active material during its conversion reaction and with PPy to offer the active material with fast electron transfer. Thus, the CoSe2@PPy-S composite cathode is reported to have improved capacity, rate performance, and cycle life.

General comment:

This research reports the common mainstream method in the lithium-sulfur batteries. The improved performance and analytical data of the nanocomposite would extend the relative research filed. Minor revisions are suggested to provide the necessary data and parameters. Hope the authors find the comment useful.

Comments:

(1) In the experimental section, the preparation of ZIF-67 and hollow CoSe2 samples is suggested to report the exact weights of the raw material in the methanol solution and to give the stirring time. The propose of a 24h resting time is suggested to be explained. The following step is suggested to report the exact weights of ZIF-67 and selenium powders. In the Electrochemical characterization of materials, the sulfur loading and content are necessary to be reported. The amount of electrolyte is necessary to be reported as the electrolyte-to-sulfur ratio.

[Suggestion] Please polish the experimental section and report the necessary experimental steps and information.

(2) In the TGA analysis, the weight loss is suggested to be analyzed carefully. First, the analysis possibly tells the major weight loss at 320C for sulfur. The following weight loss might result from other material in the nanocomposite. Thus, the weight loss from 320-500C is suggested to be identified, and the sulfur content is necessary to be recalculated. Second, to address this comment, TGA of the references sample of sulfur and CoSe2@PPy might be helpful.

[Suggestion] Please check again the TGA data and correct the content of sulfur in the composite and in the cathode. Reference data are suggested.

(3) In the analysis section, the content of Co is suggested to be analyzed. The XPS peak of Co is weak. The initial capacity loss of the cathode is large, which needs to be discussed. The discharge/charge efficiency of the cell with the CoSe2@PPy-S composite cathode is low and keeps decreasing. This needs additional discussion.

[Suggestion] Please analyze the content of Co to support the XPS analysis. Please explain the initial capacity loss and the discharge/charge efficiency of the cells.

(4) The charge contribution ratio of capacitance and diffusion is questionable. According to the CV data and discharge/charge curves, the reaction is battery electrode in charge. Why should the capacitance and diffusion be considered, especially the CV area shows no capacitance behavior? Then, in the battery CV data, the capacitance comes from what material should be clarified. Moreover, the calculation is based on peak 1 and peak 2. However, peak 2 is not the peak current. Why not use and include the peak 3 in the calculation? In addition, the CV curves changes. The peak current should be replaced by the integrated CV area for the calculation.

[Suggestion] Please consider the electrode behavior carefully.

(5) The engineering design for an advanced sulfur cathode and its application for the lithium-sulfur batteries are suggested. Some recent works showing advanced design of cathode are suggested to be added in supporting the introduction (Chemical Engineering Journal 2022, 429, 132257; Advanced Materials 2022, 34, 2108835; Journal of Alloys and Compounds 2022, 907, 164396)

[Suggestion] Please update the citation list.

Author Response

Dear reviewer:

Thank you for your decision and constructive comments on my manuscript. We have carefully considered the suggestions of the Reviewers. We have tried our best to improve and made some changes to the manuscript.

The yellow part has been revised according to your comments. Revision notes, point-to-point, are given as follows:

(1) In the experimental section, the preparation of ZIF-67 and hollow CoSe2 samples is suggested to report the exact weights of the raw material in the methanol solution and to give the stirring time. The propose of a 24h resting time is suggested to be explained. The following step is suggested to report the exact weights of ZIF-67 and selenium powders. In the Electrochemical characterization of materials, the sulfur loading and content are necessary to be reported. The amount of electrolyte is necessary to be reported as the electrolyte-to-sulfur ratio.

[Suggestion] Please polish the experimental section and report the necessary experimental steps and information.

Response: Thank you very much for your careful review and constructive suggestions with regard to our manuscript.

In the previous submission, we put the experimental parts such as sample preparation as well as test conditions in the Supplementary material. Now we've put them in the manuscript and refined the details. The ratio of ZIF-67 and selenium powders, the ratio of electrolyte-to-sulfur, and other parameters are listed in the revised manuscript.

(2) In the TGA analysis, the weight loss is suggested to be analyzed carefully. First, the analysis possibly tells the major weight loss at 320C for sulfur. The following weight loss might result from other material in the nanocomposite. Thus, the weight loss from 320-500C is suggested to be identified, and the sulfur content is necessary to be recalculated. Second, to address this comment, TGA of the references sample of sulfur and CoSe2@PPy might be helpful.

[Suggestion] Please check again the TGA data and correct the content of sulfur in the composite and in the cathode. Reference data are suggested.

Response: Thank you very much for the suggestions. In this paper, we mistakenly regard the final weight loss as the weight loss at 300℃. This error has been corrected. Most of the literature also used selenide heating to 600℃ for constant weight calculation[1, 2]. We conducted the TGA test of CoSe2-S for comparison. According to the thermogravimetric curve of the material, the loss within this temperature range is mainly due to the evaporation of sulfur, which accounts for about 70%. It is consistent with the mass proportion of sulfur in the substrate. Weight loss temperature of CoSe2-S is significantly lower than that of CoSe2@PPy-S. The thermal stability of CoSe2@PPy-S has been improved. This indicates that CoSe2@PPy-S has better structural stability and the sulfur it contains is not easily evaporated.

[1] Han Su, Longquan Lu, Mingzhi Yang, et al. Decorating CoSe2 on N-doped carbon nanotubes as catalysts and efficient polysulfides traps for Li-S batteries, Chemical Engineering Journal, 2022, 429, 132167

[2] Bingshu Guo, Qianru Ma, Longcheng Zhang, et al. Yolk-shell porous carbon spheres@CoSe2 nanosheets as multilayer defenses system of polysulfide for advanced Li-S batteries, Chemical Engineering Journal, 2021, 413, 127521

[3]   Gao L., Huang N., Wang J. , et al. Fabrication of Polypyrrole Coated Cobalt Manganate Porous Nanocubes by a Facile Template Precipitation and Annealing Method for Lithium–Sulfur Batteries. Journal of Alloys and Compounds, 2021, 885, 161350.

 (3) In the analysis section, the content of Co is suggested to be analyzed. The XPS peak of Co is weak. The initial capacity loss of the cathode is large, which needs to be discussed. The discharge/charge efficiency of the cell with the CoSe2@PPy-S composite cathode is low and keeps decreasing. This needs additional discussion.

[Suggestion] Please analyze the content of Co to support the XPS analysis. Please explain the initial capacity loss and the discharge/charge efficiency of the cells.

Response: Thank you very much for the suggestions. The content of Co element (7.31%) in CoSe2@PPy-S composite was determined by elemental energy spectrum analysis.

In Figure 6(c), it shows cyclic properties of bare sulfur, CoSe2-S and CoSe2@PPy-S composites at 0.2 C. The capacity loss of the cathode is large in the first three cycles.  Because the charge-discharge rate of the first 3 cycles was 0.05 C. The main purpose of this step was conducted to activate the electrode.

The reason of discharge/charge efficiency keeps decreasing is the increase of electrolyte viscosity and the decrease of ionic conductivity due to the dissolution and diffusion of polysulfide during the cycle of lithium sulfur battery.

(4) The charge contribution ratio of capacitance and diffusion is questionable. According to the CV data and discharge/charge curves, the reaction is battery electrode in charge. Why should the capacitance and diffusion be considered, especially the CV area shows no capacitance behavior? Then, in the battery CV data, the capacitance comes from what material should be clarified. Moreover, the calculation is based on peak 1 and peak 2. However, peak 2 is not the peak current. Why not use and include the peak 3 in the calculation? In addition, the CV curves changes. The peak current should be replaced by the integrated CV area for the calculation.

[Suggestion] Please consider the electrode behavior carefully.

Response: Thank you very much for the suggestions. Here we have a writing error. We miswrote K3 as K2. In order to further analyze the electrode dynamics information of CoSe2@PPy-S during the charging and discharging process and its contribution to the capacity. Corresponding b value would be fitting from CV curves of CoSe2@PPy-S electrode. The b values of peak 1 and peak 3 are 0.765, and 0.654 respectively, which are a mixture of capacitive type and diffusion type. With the increase of scanning rate, the contribution ratio of pseudocapacitance increases gradually. This shows that the contribution capacity of the pseudocapacitor comes from a rapid charge-discharge process, and the influence of the pseudocapacitor is weaker than that of the electrochemical reaction at higher charge-discharge current [1].

Reference: [1] Lu P, Sun Y, Xiang H, et al. 3D Amorphous Carbon with Controlled Porous and Disordered Structures as a High-Rate Anode Material for Sodium-Ion Batteries [J]. Advanced Energy Materials, 2018, 8(8):1702434.

(5) The engineering design for an advanced sulfur cathode and its application for the lithium-sulfur batteries are suggested. Some recent works showing advanced design of cathode are suggested to be added in supporting the introduction (Chemical Engineering Journal 2022, 429, 132257; Advanced Materials 2022, 34, 2108835; Journal of Alloys and Compounds 2022, 907, 164396)

[Suggestion] Please update the citation list.

Response: Thank you very much for the suggestions. We have updated the references and cited the above references.
